# Diosmin and Hesperidin Have a Protective Effect in Diabetic Neuropathy via the FGF21 and Galectin-3 Pathway

**DOI:** 10.3390/medicina60101580

**Published:** 2024-09-26

**Authors:** Birzat Emre Gölboyu, Mümin Alper Erdoğan, Mehmet Ali Çoşar, Ezgi Balıkoğlu, Oytun Erbaş

**Affiliations:** 1Department of Anesthesiology and Reanimation, Izmir Katip Celebi University Ataturk Training and Research Hospital, Izmir 35000, Türkiye; 2Department of Physiology, Faculty of Medicine, Izmir Katip Celebi University, Izmir 35000, Türkiye; 3Department of Anesthesiology and Reanimation, Dr. Suat Seren Gögüs Hastalıkları Hastanesi, Izmir 35000, Türkiye; 4Department of Physiology, Demiroğlu Bilim University, Istanbul 34000, Türkiye

**Keywords:** diabetic neuropathy, diosmin, FGF21, galectin-3, hesperidin, oxidative stress, nerve regeneration

## Abstract

*Background and Objectives*: This study aimed to investigate the protective effect of diosmin and hesperidin in diabetic neuropathy using a rat model, focusing on their impact on nerve regeneration through the fibroblast growth factor 21 (FGF21) and galectin-3 (gal3) pathway. *Materials and Methods:* Forty adult male Wistar rats were used in this study. Diabetes was induced using streptozotocin (STZ), and the rats were divided into control, diabetes and saline-treated, diabetes and diosmin + hesperidin (150 mg/kg) treated, and diabetes and diosmin + hesperidin (300 mg/kg) treated groups. Electromyography (EMG) and inclined plane testing were performed to assess nerve function and motor performance. Sciatic nerve sections were examined histopathologically. Plasma levels of FGF21, galectin-3, and malondialdehyde (MDA) were measured as markers of oxidative stress and inflammation. *Results:* Diabetic rats treated with saline displayed reduced nerve conduction parameters and impaired motor performance compared to controls. Treatment with diosmin and hesperidin significantly improved compound muscle action potential (CMAP) amplitude, distal latency, and motor performance in a dose-dependent manner. Histopathological examination revealed decreased perineural thickness in treated groups. Additionally, treatment with diosmin and hesperidin resulted in increased plasma FGF21 levels and reduced plasma levels of galectin-3 and MDA, indicating decreased oxidative stress and inflammation. *Conclusions:* Diosmin and hesperidin exhibited protective effects in diabetic neuropathy by promoting nerve regeneration, enhancing nerve conduction, and improving motor performance. These effects were associated with modulation of the FGF21 and galectin-3 pathway. These findings suggest that diosmin and hesperidin may hold potential as adjunctive therapies for diabetic neuropathy.

## 1. Introduction

Diabetic neuropathic pain (DNP), a microvascular complication of diabetes mellitus (DM), often results in significant clinical morbidity. The clinical manifestation of DNP encompasses various pain modalities (such as burning, tingling, and sharp and lancinating sensations), pain triggered by normally nonpainful stimuli (hyperalgesia and allodynia), and atypical sensations (such as intense cold or swelling in the legs). The pathogenesis of DNP involves a complex interplay of factors due to the multifaceted nature of the disease. Persistent hyperglycemia remains the foremost contributor, leading to metabolic alterations that contribute to DNP. Effectively managing it remains a substantial clinical challenge due to a limited understanding of its underlying causes. Current therapeutic approaches primarily focus on addressing the symptoms of neuropathic pain and encompass a range of treatments, including antidepressants, anticonvulsants, gamma-aminobutyric acid (GABA) agonists, opioid medications, topical capsaicin, and local anesthetic ointments. It is important to note, however, that not all patients respond effectively to these treatments, and that these medications are often linked to significant adverse effects. To enhance the management of DNP, there is a pressing need for innovative medications that can efficiently and safely alleviate neuropathic pain symptoms. Such drugs are essential for enhancing the treatment options available to DNP patients and improving their quality of life [1,2].

Recent trends in complementary and alternative medicine (CAM) have garnered increasing attention due to their potential role in the management of diabetes and its associated metabolic complications. Comprehensive research endeavors have delved into CAM modalities, yielding evidence that these approaches can serve as adjunctive therapies by proficiently ameliorating glycemic control, HbA1c levels, and dyslipidemia. Additionally, these modalities exhibit the capacity to mitigate diabetes-related complications, including diabetic neuropathy, thereby fostering an improved quality of life for individuals afflicted with type 2 diabetes (T2DM). A salient illustration pertains to hesperidin and diosmin flavonoids derived from botanical sources, renowned for their multifaceted attributes spanning antihyperlipidemic, anti-inflammatory, analgesic, antioxidant, antidiabetic, and antihypertensive properties. Their notable tolerability and safety profiles have been extensively documented [3]. Moreover, both hesperidin and diosmin, whether used individually or in synergy, have demonstrated favorable effects on diabetic neuropathy and dyslipidemia in animal models [4]. Plant-derived flavonoids, particularly diosmin, have garnered attention for their medicinal properties. Diosmin, a flavone glycoside, exhibits a wide range of biological activities including antioxidant, antihyperglycemic, anti-inflammatory, and antiulcer effects; it also has therapeutic potential, and key biological properties, such as anticancer and neuroprotective effects [5].

Fibroblast growth factor 21 (FGF21) constitutes an endogenous hormone expressed in the liver, adipose tissue, and pancreas, elicited by conditions of starvation-induced stress or certain pathological states. FGF21 encompasses diverse physiological functions, including hypoglycemic, anti-inflammatory, and antioxidative activities [6]. Galectin-3 (gal3) is a member of the β-galactoside-binding lectin family, representing a versatile protein with diverse biological roles. These functions encompass activities such as promoting cancer cell proliferation, facilitating invasion, and stimulating angiogenesis [7].

The main objective of the present study was to investigate the efficacy of diosmin and hesperidin in promoting nerve regeneration in a rat model of diabetic neuropathy involving the sciatic nerve.

## 2. Materials and Methods

### 2.1. Animals

A total of 40 adult male Wistar rats, weighing between 200 and 210 g, were included in this study. The rats were housed in standard cages under controlled conditions, with a 12 h light/dark cycle and a room temperature of 22 ± 2 °C. They were provided with ad libitum access to a standard pellet diet and tap water throughout the study period. The experimental protocol involving the animals was approved by the Institutional Animal Care and Ethical Committee of the University of Science (Ethical Number: 1523034701, Approval date: 3 October 2022). Chemicals and reagents, unless otherwise specified, were sourced from Sigma-Aldrich Inc. (Sigma-Aldrich, Inc.; Saint Louis, MO, USA). Institutional and national standards for the care and use of laboratory animals were followed.

### 2.2. Experimental Protocol

Thirty rats were injected intraperitoneally (IP) with a single dose of streptozotocin (STZ) (Sigma-Aldrich, Inc.; Saint Louis, MO, USA) (60 mg/kg in 0.9% NaCl, pH 4.0 adjusted with 0.2 M sodium citrate). Ten rats were allocated to the control group (*n* = 10, control group) and were not administered any chemicals. Normal blood glucose levels in the control group were <120 mg/dL. After 24 h, DM was confirmed by measuring the blood glucose levels with glucose oxidase reagent strips (Boehringer-Mannheim, Indianapolis, USA). The diabetic rat had blood glucose levels of 250 mg/dl or higher. 

Subsequently, the 30 diabetic rats were randomly divided into three groups: Group 1 rats (*n* = 10, diabetes and saline group) were administered 1 mL/kg of saline treatment via oral gavage. Group 2 rats (*n* = 10, diabetes and diosmin + hesperidin treatment group) received diosmin at 135 mg/kg/day and hesperidin at 15 mg/kg/day (total 150 mg/kg) through oral gavage. These compounds were administered in the form of Daflon (500 mg, SERVIER). Group 3 rats (*n* = 10, diosmin + hesperidin treatment group) were treated with diosmin at 270 mg/kg/day and hesperidin at 30 mg/kg/day (total 300 mg/kg) through oral gavage over a duration of 4 weeks. At the end of the 4-week treatment period, following the administration of diosmin and hesperidin in the respective groups, electromyography (EMG) and inclined plane testing were conducted. The animals were then euthanized, blood samples were obtained by cardiac puncture for biochemical analysis, and the sciatic nerve was extracted for histological investigation.

EMG was procured three times from the right sciatic nerve and stimulated supramaximally (intensity 10 V, duration 0.05 ms, frequency 1 Hz, in the range of 0.5–5000 Hz, 40 kHz/sec sampling rate) from the Achilles tendon using a Biopac bipolar subcutaneous needle stimulation electrode (Biopac Systems, Inc., Santa Clara, CA, USA). In the interosseous muscle, unipolar needle electrodes were used to collect compound muscle action potentials (CMAPs) and variations in motor nerve conduction velocity (NCV). Biopac Student Lab Pro version 3.6.7 (Biopac Systems, Inc., Goleta, CA, USA) was used to analyze the data, distal latency, duration, and amplitude of the CMAP, which served as the parameters (Figure 1).

#### Components of EMG Recording System

Stimulating electrodes: The stimulating electrodes are placed near the sciatic notch of the rat. These electrodes are responsible for delivering a controlled electrical stimulus to the sciatic nerve, inducing a response that can be measured further down the nerve.

Recording electrodes: Recording electrodes are placed distally, typically in the muscles of the hind paw (interosseous muscle), to detect the compound muscle action potentials (CMAPs) resulting from the nerve stimulation. 

Grounding electrode: A grounding electrode is attached to the rat’s tail to prevent electrical interference during the recording process.

EMG recording device: The recording device shown at the top of the figure is used to monitor and record the EMG signals. It is connected to the stimulating and recording electrodes via leads. The device displays the output in real-time, allowing for immediate observation of nerve conduction parameters. 

Temperature control: A temperature control probe might be seen, which is often placed to monitor the body temperature of the rat during the experiment, ensuring that the temperature remains stable, as fluctuations can affect nerve conduction results.

Within the EMG recordings, the rectal temperatures of the rats were measured using a rectal probe (HP Viridia 24-C; Hewlett-Packard Company, Palo Alto, CA, USA), and each rat’s temperature was maintained between 36 and 37 °C using a heating pad. 

### 2.3. Histopathological Examination of the Sciatic Nerve

The sciatic nerve sections were fixed in formalin and processed for histopathological examination. Sections of 4 μm thickness were stained with hematoxylin and eosin (H and E) for structural analysis. The perineural thickness of the sciatic nerve was quantified using an Olympus C-5050 digital camera mounted on an Olympus BX51 microscope. Measurements were taken at three distinct points per section to capture variability across different regions of the nerve. For each group, nerve samples from 10 animals (*n* = 10) were analyzed, and measurements were performed in triplicate for each sample to ensure accuracy and consistency. The data obtained from these measurements were averaged for each sample.

Statistical analysis was performed using one-way analysis of variance (ANOVA) to compare perineural thickness between groups, followed by Tukey’s post hoc test for multiple comparisons. A significance level of *p* < 0.05 was considered statistically significant. Data are presented as mean ± standard error of the mean (SEM).

### 2.4. Inclined Plane Test

The evaluation of motor performance in rats was conducted using an inclined plane test, modified from a sliding apparatus, as described previously [8]. The test was conducted one month after the induction of STZ-induced diabetes. The sliding apparatus had a 50 cm × 30 cm stainless steel plane. The maximum angle was then determined at the moment just when a limb of the rat slipped in order to maintain body position. The test was performed three times for each head position and averaged. Each trial was performed after a minute interval (Figure 2).

### 2.5. Evaluation of Lipid Peroxidation

The assessment of lipid peroxidation involved the determination of malondialdehyde (MDA) levels in plasma samples, quantified as thiobarbituric acid reactive substances (TBARSs). The procedure entailed the addition of trichloroacetic acid and TBARS reagent to the plasma samples, followed by incubation at 100 °C for 60 min. After cooling on ice, the samples were centrifuged at 3000 rpm for 20 min and the absorbance of the supernatant was read at 535 nm. MDA levels were expressed as nanomoles (nM), and tetraethoxypropane was used for calibration.

### 2.6. Measurement of Plasma FGF21 and Galectin-3 Level 

Plasma levels of FGF21 and galectin-3 were measured using commercially available enzyme-linked immunosorbent assay (ELISA) kits (BD Bioscience, San Jose, CA, USA). The levels of FGF21 and galectin-3 were expressed in pg/mL.

### 2.7. Statistical Analysis

SPSS version 22.0 (IBM Corp., Armonk, NY, USA) for Windows was used for the data analysis. Student’s *t*-test and analysis of variance (ANOVA) were used to compare the parametric variables, while the nonparametric variables were compared using the Mann–Whitney U test. Results were reported as the mean standard deviation of the mean (SEM). A *p*-value of less than 0.05 was considered statistically significant. *p* < 0.001 was accepted as statistically highly significant.

## 3. Results

### 3.1. EMG Recordings 

Figure 3 displays EMG recordings from distinct experimental groups, including the control group (a), diabetic group treated with saline (b), diabetic group treated with diosmin + hesperidin at 150 mg/kg (c), and diabetic group treated with diosmin + hesperidin at 300 mg/kg (d). In the group subjected to diabetes and saline treatment, there was a significant reduction in compound muscle action potential (CMAP) amplitude in comparison to the control group (*p* < 0.05). Additionally, distal latency was significantly prolonged when contrasted with the control group (*p* < 0.05). Conversely, the diabetes and diosmin + hesperidin 150 mg/kg treatment group, exhibited a significantly heightened CMAP amplitude compared to the diabetes and saline treatment group (*p* < 0.05). The distal latency was also significantly reduced compared to the diabetes and saline treatment group (*p* < 0.05). In the diabetes and diosmin + hesperidin 300 mg/kg treatment group, the CMAP amplitude was even higher than the diabetes and diosmin + hesperidin 150 mg/kg treatment group (*p* < 0.01). Concurrently, distal latency was significantly decreased when compared to the diabetes and saline treatment group (*p* < 0.05) (Table 1).

### 3.2. Histological Analysis of Sciatic Nerve Sections

In this section, we present a histological analysis of longitudinal sections of the sciatic nerve. This orientation was chosen to provide a detailed view of the nerve fibers along their length, allowing for the assessment of features such as axonal integrity, myelin sheath thickness, and perineurium structure. In longitudinal sections, the nerve fibers (axons) appear as elongated structures running parallel to the section plane, while the perineurium surrounds these fibers in a continuous layer, conducted at a magnification of x40 with H and E staining; distinct observations were made across different experimental groups. In the control group (a), the perineurium and perineural thickness were apparent, accompanied by well-defined axons. Conversely, the group treated with diabetes mellitus (DM) and saline (b) exhibited an increase in perineural thickness. Notably, in groups treated with DM and diosmin + hesperidin, at doses of 150 mg/kg (c) and 300 mg/kg (d), a discernible decrease in perineural thickness was observed (Figure 4.).

### 3.3. Inclined Plane Test

The maximum angle observed during the inclined plane test (in degrees) was markedly diminished in the diabetes and saline treatment group in contrast to the control group (* *p* < 0.05). However, both the diabetes and diosmin + hesperidin 150 mg/kg treatment group (# *p* < 0.05) and the diabetes and diosmin + hesperidin 300 mg/kg treatment group (# *p* < 0.05) demonstrated a significant increase in this angle relative to the diabetes and saline treatment group. Plasma glucose levels (mg/dl) were significantly increased in the diabetes and saline treatment group compared to the control group (* *p* < 0.05). In contrast, no significant differences were discerned in the diabetes and diosmin + hesperidin 150 mg/kg treatment group and the diabetes and diosmin + hesperidin 300 mg/kg treatment group relative to the diabetes and saline treatment group (Table 2).

### 3.4. Perineural Thickness and Biomarkers

The evaluation of perineural thickness uncovered a marked increase in the diabetic group subjected to saline treatment. However, both doses of diosmin plus hesperidin significantly reduced this thickness. Notably, plasma levels of fibroblast growth factor 21 (FGF21) were also significantly elevated in the diabetic group treated with saline; nevertheless, administration of diosmin plus hesperidin at either dose induced a further substantial augmentation in FGF21 levels. Similarly, plasma levels of galectin-3 and malondialdehyde (MDA), markers indicative of inflammation and oxidative stress, respectively, demonstrated a significant escalation in the diabetic group treated with saline. Conversely, treatment with diosmin plus hesperidin at either dosage conferred a considerable reduction in these levels (Table 3).

## 4. Discussion

In light of the intricate challenges posed by DNP and the limited efficacy of current therapeutic approaches, there is a growing impetus to explore alternative strategies that can provide more comprehensive relief from the debilitating symptoms associated with this condition. As DNP continues to impact the quality of life of individuals with DM, it is crucial to pursue novel interventions that target underlying pathophysiological mechanisms. CAM modalities have gained traction as potential adjunctive therapies for diabetes and its complications, offering multifaceted benefits encompassing glycemic control, anti-inflammatory properties, and antioxidant effects. Among these modalities, diosmin and hesperidin have emerged as promising candidates due to their diverse pharmacological attributes. Additionally, FGF21 and galectin-3 (gal3) play notable roles in neuroinflammation and oxidative stress regulation, presenting intriguing avenues for therapeutic exploration. This study seeks to delve into the potential of diosmin and hesperidin in promoting nerve regeneration in a diabetic neuropathy rat model while also examining their influence on FGF21 and gal3 levels. By investigating these interactions, the research aims to contribute valuable insights into the development of innovative strategies for managing DNP and its associated complications.

The presence of flavonoids in the Citrus genus was first identified in the late nineteenth century, with hesperidin being a notable discovery. Since then, 44 naturally occurring flavonoids in citrus have been identified. These flavonoids are found in various citrus fruits, such as bergamots, grapefruits, lemons, limes, mandarins, oranges, and pomelos. The range of citrus flavonoids includes compounds such as flavones, flavanones, flavonols, isoflavones, anthocyanidins, and flavanols. Some are unique to the Citrus genus, such as polymethoxyflavones (PMFs), while others are specific to certain varieties [9].

Diosmin, originally isolated in 1925 from Scrophularia nodosa Linn., a perennial herbaceous plant of the Scrophulariaceae family, has exhibited a diverse range of therapeutic potential since its discovery. Initially employed as a treatment for inflammatory disorders, diosmin’s clinical applications have evolved over time. Currently, it is primarily recognized for its efficacy in addressing chronic venous insufficiency and hemorrhoids [3]. However, recent research has unveiled its multifaceted protective effects in the context of diabetic neuropathy. In a study conducted on type 2 diabetic rats fed a high-fat diet, diosmin demonstrated significant safeguarding effects against a spectrum of factors associated with diabetic neuropathy, including biochemical aberrations, behavioral manifestations, and oxidative stress markers. Notably, this flavonoid exhibited the capacity to enhance the pain threshold in tests assessing thermal hyperalgesia and tail-flick responses, concurrently ameliorating motor function in diabetic rats. An additional facet of diosmin’s protective attributes manifested in its ability to counteract oxidative stress by attenuating lipid peroxidation markers, specifically MDA and nitric oxide (NO), while concurrently reinforcing the activity of crucial antioxidant enzymes such as superoxide dismutase (SOD) and reduced glutathione (GSH). These findings underscore the potential of diosmin to mitigate the early stages of diabetic neuropathy in this experimental rat model [10].

Moreover, diosmin exhibited a role in glycemic regulation, as evidenced by its capacity to reduce elevated plasma glucose levels and enhance hepatic glycogen content in diabetic rats. Notably, this therapeutic effect was attributed to its interaction with the I-2R receptor, fostering metabolic equilibrium and contributing to the reduction of both blood glucose and lipid levels in diabetic rats. Importantly, diosmin’s therapeutic effects were accomplished without causing significant alterations in body weight, food intake, or plasma insulin levels [11,12]. Furthermore, its positive impact extends to behavioral domains, encompassing antinociceptive responses, locomotor activity, and the regulation of nociceptive biomarkers linked to diabetes-induced neuropathy [10].

One study investigated the potential therapeutic effects of diosmin on diabetic neuropathy in type 2 diabetic rats. The study utilized a rat model of type 2 diabetes induced by streptozotocin and a high-fat diet. Diosmin treatment over a four-week period showed significant improvements in various parameters. It reduced blood glucose levels and insulin resistance, counteracted cold allodynia (sensitivity to cold stimuli), and improved walking function in diabetic rats. Furthermore, diosmin treatment led to a reduction in oxidative stress indicators and an elevation in antioxidant enzyme activity. These findings suggest that diosmin may have a beneficial impact on diabetic neuropathy and related complications in this animal model [13].

Hesperidin, a flavanone glycoside abundantly present in citrus fruits such as oranges, tangerines, lemons, and grapefruits, is a significant nonessential nutrient for humans [12]. This citrus flavanone is widely utilized as a dietary supplement, both independently and in combination with other bioflavonoids. Its applications encompass the treatment or prevention of vascular system disruptions (reducing capillary permeability), serving as an anti-inflammatory, antioxidant, or anticarcinogenic herbal remedy [14]. Many of its medicinal attributes are attributed to its capacity to regulate pro-inflammatory cytokines, such as TNF-α, IL-1β, and IL-6, thereby mitigating inflammation and oxidative stress in various animal models of inflammatory reactions [4]. Mahmoud et al. highlighted hesperidin’s potential in attenuating hyperglycemia-induced oxidative stress and suppressing the production of pro-inflammatory cytokines, such as TNF-α and IL-6, in type 2 diabetic rats induced by a high-fat diet and streptozotocin (STZ). These effects manifested in both physical and behavioral outcomes, including reduced immobility time in a forced swimming test for diabetic rats [15]. Visnagri et al. delved into hesperidin’s influence on diabetic neuropathic pain in rats. Treatment with hesperidin was found to counteract the reduction in motor and sensory nerve conduction velocity induced by diabetes. This is significant, as impaired neural conduction velocity can impact sensory, nociceptive, and motor responses. Hesperidin also ameliorated sensory responses by mitigating heightened mechanical and thermal hypersensitivity in diabetic rats. Moreover, it improved several diabetic biochemical markers, including blood glucose levels, total cholesterol (TC), and serum triglycerides (TGs) while increasing plasma insulin concentration and exhibiting positive effects on hemodynamic variables crucial for diabetes treatment and associated cardiovascular complications. Hesperidin demonstrated neural protective effects, which are evident from decreased neutrophil and macrophage infiltration in the sciatic nerve and decreased mRNA expression of neural TNF-α and IL-1β, two key pro-inflammatory cytokines in diabetes progression. Furthermore, it restored the sciatic nerve’s architecture distortion caused by STZ-induced necrosis, edema, and fiber congestion [16].

Mahmoud et al. provided further insights into hesperidin’s mechanism of action as a natural antidiabetic agent using HFD/STZ-induced type 2 diabetic rats and in vitro studies. The oral administration of hesperidin was observed to lower fasting glucose levels, alleviate insulin resistance in diabetic rats, and enhance insulin release in isolated pancreatic islets. It also normalized the activities of metabolic enzymes, including glucose-6-phosphatase, glycogen phosphorylase, and fructose-1,6-bisphosphatase. This bioflavonoid increased glucose uptake in isolated pancreatic cells. The antidiabetic effects of hesperidin were predominantly attributed to its capacity to augment mRNA and the protein expression of GLUT4 in adipose tissue while also reducing intestinal glucose absorption [17].

In one study, a comprehensive exploration was undertaken to assess the effects and underlying mechanisms of hesperidin on a spectrum of pathophysiological parameters associated with diabetic retinopathy. The investigation employed retinal RGC-5 ganglion cells as the experimental model. It is well acknowledged within the scientific community that oxidative stress constitutes a pivotal element in the pathogenesis of diabetes, characterized by heightened concentrations of reactive oxygen species (ROS) and the biomarker of lipid peroxidation, MDA, juxtaposed with a decline in the enzymatic activity of antioxidant defenses. Upon exposure to elevated glucose concentrations, RGC-5 cells demonstrated a pronounced decrease in intracellular ROS, MDA, and protein carbonyl content upon hesperidin administration. Furthermore, the compromised activities of pivotal antioxidant enzymes, such as superoxide dismutase (SOD), glutathione peroxidase (GPx), and catalase (CAT), were effectively restored by hesperidin treatment. The research findings also illuminated the potential of hesperidin to effectively counteract the upregulation of Bax, a pivotal regulator of apoptotic pathways evoked by elevated glucose levels. Simultaneously, hesperidin treatment led to a reduction in Bcl-2 levels in RGC-5 cells subjected to hyperglycemic conditions. Concomitantly, the activities of caspase-9 and caspase-3, both integral to apoptotic cascades, were attenuated by hesperidin. This attenuation culminated in a shift in the Bax/Bcl-2 ratio toward a mitigated apoptotic propensity. Furthermore, the restoration of mitochondrial function, underscored by enhanced mitochondrial integrity and activity, provided evidence of hesperidin’s potential to safeguard cells against apoptosis triggered by high glucose levels [13].

In one investigative study, the focus was directed toward exploring potential therapeutic interventions for traumatic facial nerve injuries through an experimental animal model. The treatments under scrutiny included methylprednisolone, hyperbaric oxygen, and a combined administration of hesperidin and diosmin. Rats afflicted with facial nerve injuries were segregated into five distinct groups, negative control, operation-only, corticosteroid, hyperbaric oxygen, and hesperidin–diosmin. The assessment encompassed the evaluation of blink reflex responsiveness as well as the microscopic analysis of facial nerve samples. The hyperbaric oxygen group demonstrated a discernible reduction in axonal degeneration and vascular congestion when juxtaposed with the operation-only and corticosteroid groups. This group also exhibited elevated myelin sheath thickness. The hesperidin–diosmin group similarly displayed mitigated axonal degeneration and vascular congestion. Notably, no statistically significant variations in functional recovery were observed across the different treatment groups. The outcomes underscored the constructive influence of both hyperbaric oxygen and hesperidin–diosmin treatments on the process of facial nerve regeneration within the experimental framework. Nevertheless, further in-depth investigation is imperative to ascertain the full scope and efficacy of these interventions in the context of traumatic nerve injuries [18].

In another study, investigators endeavored to elucidate the antihyperalgesic properties of hesperidin, both in isolation and in conjunction with diosmin, utilizing a rat model of neuropathic pain. The examination encompassed an evaluation of mechanical and thermal hypersensitivity in rats afflicted with neuropathic pain, followed by the administration of varying dosages of hesperidin, either as a standalone treatment or in combination with diosmin. The combination of hesperidin and diosmin showed enhanced pain relief. The study also revealed that the antihyperalgesic effects involve central nervous system activity influenced by D2 receptors, GABAA receptors, and opioid receptors, but not 5-HT1A receptors. Both compounds were detected in the brain samples. This suggests that hesperidin and diosmin could be potential agents for neuropathic pain relief and that their combination might have a synergistic effect [4].

In a paper investigating the antioxidant protective effects of diosmin and hesperidin, natural citrus flavones, when combined, protected against heavy metal-induced damage and oxidative stress in Wistar albino rats. Rats were given oral doses of diosmin and hesperidin (200 mg/kg and 100 mg/kg, respectively) for a month before exposure to heavy metals. The combination effectively countered the harmful effects of heavy metals, including restoring normal gene expression (P53 and caspase-3), improving antioxidant levels, reducing lipid peroxidation and DNA damage, and preventing tissue damage in the liver. Importantly, the diosmin and hesperidin combination showed protective effects without causing adverse effects on its own. Overall, the study suggests that this combination could be a promising strategy for mitigating heavy metal-induced damage and oxidative stress in rats [19].

Within the framework of this study, the primary objective was to elucidate the impacts of FGF21 on neurodegenerative processes in the context of aging and diabetes in murine models. The researchers administered FGF21 or metformin to mice daily for 4 to 6 months and then further studied the mechanisms in SH-SY5Y cells. The study explored relevant gene expression, protein levels, histological changes, and biochemical markers. FGF21 inhibited neuron loss in diabetic and aging mice, as demonstrated by the NeuN protein levels. Histological staining indicated that FGF21 reduced cellular damage and edema around specific areas of the brain associated with memory (dentate gyrus and CA3). FGF21 also suppressed the aggregation of tau and β-amyloid1−42, which is linked to neuron apoptosis and neurodegenerative processes. Importantly, FGF21 administration reduced neuroinflammation by decreasing the expression of Iba1, NF-κB, IL6, and IL8 while enhancing antioxidant enzyme levels in both aging and diabetic mice. In addition, FGF21 treatment increased the phosphorylation of AKT and AMPKα, which are key signaling pathways associated with cell survival and energy regulation. In in vitro experiments using SH-SY5Y cells, FGF21 inhibited tau and β-amyloid1−42 aggregation induced by lipopolysaccharide (LPS), a molecule that triggers inflammation. This effect was associated with decreased NF-κB expression and increased phosphorylation of AKT and AMPKα. FGF21 demonstrated neuroprotective effects by reducing neuroinflammation and oxidative stress, likely by regulating the NF-κB pathway and the AMPKα/AKT pathway. This mechanism enhanced mitochondrial protection in neurons, suggesting that FGF21 has potential therapeutic implications for attenuating neurodegeneration in aging and diabetic contexts [20].

This study explored the potential of FGF21 as a therapeutic strategy for enhancing nerve regeneration following PNI. Using a rat model of sciatic nerve crush injury, researchers found that administering FGF21 resulted in improved functional recovery, increased axonal regrowth, and enhanced Schwann cell proliferation. FGF21 application also reduced oxidative stress and autophagic cell death in Schwann cells. These effects were associated with the inhibition of the ERK/Nrf-2 signaling pathway, suggesting that FGF21 could be a valuable therapy for facilitating nerve regeneration and remyelination after PNI [21].

Utilizing a well-established model of PNI, this study demonstrated that the utilization of modified citrus pectin (MCP) to suppress gal3 led to a notable alleviation of neuroinflammation and the subsequent mitigation of neuropathic pain. The observations of this study encompassed multiple dimensions of gal3’s role in neuropathic pain, unraveling its contribution to inflammatory processes and pain hypersensitivity. In response to PNI, the heightened expression of gal3 was discerned, and its inhibition through MCP treatment yielded profound outcomes. By attenuating gal3 expression, MCP effectively curtailed the activation of autophagy, a process intricately linked with microglial activation and the propagation of inflammation. This intervention subsequently translated into a substantial reduction in the expression of proinflammatory mediators, representing a critical facet of neuroinflammation [22]. Notably, the heightened pain sensitivity characteristic of neuropathic pain was significantly ameliorated following gal3 inhibition, suggesting its pivotal role in driving the pathophysiological processes underlying neuropathic pain. Intriguingly, the study extended its exploration beyond the experimental setting to investigate the clinical relevance of gal3 in the context of chemotherapy-induced peripheral neuropathy (CIPN). By examining patients undergoing taxane-treated breast cancer therapy and a corresponding mouse model, the research revealed a consistent elevation in the plasma levels of gal3. This elevation emerged as a hallmark of taxane-related CIPN, implicating gal3 in the development and progression of this condition. The study illuminated a potential mechanistic link between Schwann cell-derived gal3 and the initiation of taxane-induced peripheral neuropathy, notably through its role in promoting macrophage infiltration, thereby contributing to heightened mechanical hypersensitivity [23].

## 5. Conclusions

In summary, this study underscores the potential benefits of diosmin and hesperidin in addressing the challenges posed by diabetic neuropathy. The results indicate that these compounds, administered either alone or in combination, hold promise for promoting nerve regeneration and enhancing motor function in a rat model of diabetic neuropathy. This positive impact is likely mediated through FGF21 and gal3 pathway modulation, which is pivotal in managing inflammation and oxidative stress. The observed reduction in perineural thickness, elevation of plasma FGF21 levels, and reduction in gal3 and MDA levels collectively suggest that diosmin and hesperidin could be valuable agents in mitigating the underlying pathophysiology of diabetic neuropathy. These findings not only deepen our understanding of the mechanisms driving their therapeutic effects but also highlight their potential as complementary treatments to improve the management of diabetic neuropathic pain.

## Figures and Tables

**Figure 1 medicina-60-01580-f001:**
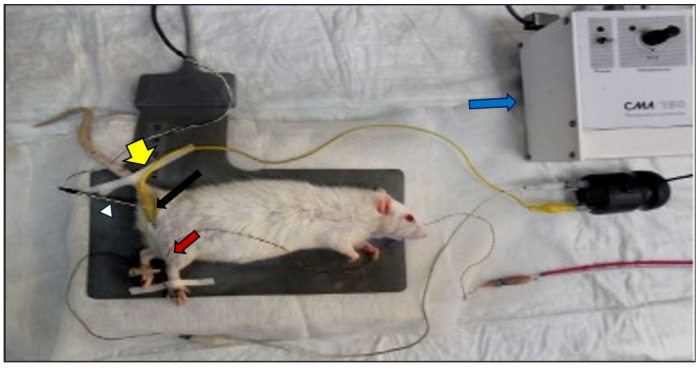
Electromyography (EMG) recording system: a Wistar rat placed in a supine position on a nonconductive surface, with electrodes strategically positioned to stimulate and record the electrical activity of the sciatic nerve. Stimulating electrodes (yellow arrow), recording electrode (red arrow), grounding electrode (white arrow head), EMG recording device (blue arrow), temperature control (black arrow).

**Figure 2 medicina-60-01580-f002:**
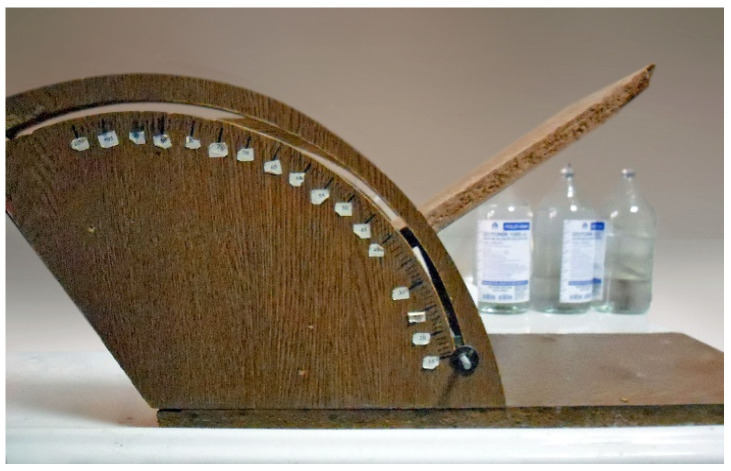
Inclined plane test system.

**Figure 3 medicina-60-01580-f003:**
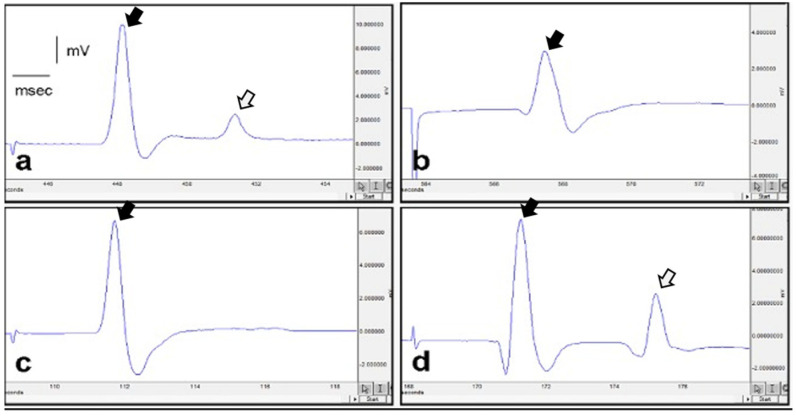
(**a**) Control group EMG, (**b**) diabetic and saline treatment EMG, (**c**) diabetic and diosmin + hesperidin 150 mg/kg EMG, (**d**) diosmin + hesperidin 300 mg/kg treatment EMG. M-wave: the initial sharp peak, representing the direct muscle response (black arrow). H-reflex (if present): a secondary, smaller peak representing the reflexive response (empty white arrow).

**Figure 4 medicina-60-01580-f004:**
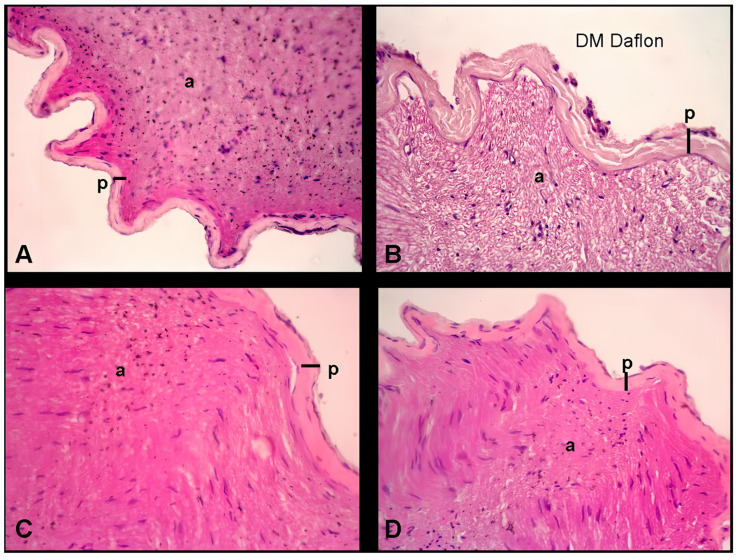
The histological sections of the sciatic nerve in different experimental groups stained with H and E (×40 magnification). These images are longitudinal sections of the nerve: (**A**) control group: the image shows a healthy sciatic nerve with a normal perineurium (p) and well-organized axons (a). (**B**) Diabetic group treated with saline: this section shows increased perineural thickness (p) compared to the control group, indicating nerve damage due to diabetes. (**C**) Diabetic group treated with diosmin + hesperidin (150 mg/kg): this section shows a decrease in perineural thickness (p) compared to the saline-treated diabetic group, suggesting a protective effect of the treatment. (**D**) Diabetic group treated with diosmin + hesperidin (300 mg/kg): similar to (**C**), this section shows further decreased perineural thickness (p), indicating a dose-dependent protective effect of the treatment. In each image, p indicates the perineurium, which is the protective sheath surrounding the nerve, and a marks the axons, the long thread-like parts of a nerve cell along which impulses are conducted.

**Table 1 medicina-60-01580-t001:** Effects of diosmin and hesperidin on CMAP amplitude and distal latency in diabetic and control groups. Data are expressed as mean ± SEM.

	Control Group	Diabetes and Saline Treatment	Diabetes and Diosmin + Hesperidin 150 mg/kg Treatment	Diabetes and Diosmin + Hesperidin 300 mg/kg Treatment
CMAP Amplitude (mV)	12.9 ± 0.6	6.6 ± 0.9 *	9.3 ± 0.4 #	11.5 ± 0.6 ##
Distal Latency (ms)	2.34 ± 0.1	3.11 ± 0.2 *	2.65 ± 0.1 #	2.61 ± 0.09 #

* *p* < 0.05 (different from control), # *p* < 0.05, ## *p* < 0.01 (different from diabetes + saline).

**Table 2 medicina-60-01580-t002:** Effects of diosmin and hesperidin on maximum angle of inclined plane test in diabetic and control groups. Data are expressed as mean ± SEM.

	Control Group	Diabetes and Saline Treatment	Diabetes and Diosmin + Hesperidin 150 mg/kg Treatment	Diabetes and Diosmin + Hesperidin 300 mg/kg Treatment
Maximum angle of Inclined plane test (degree)	87.5 ± 2.3	50.9 ± 4.1 *	69.6 ± 6.2 #	75.8 ± 3.9 #
Plasma glucose (mg/dl)	93.7 ± 5.5	398.3 ± 18.3 *	384.2 ± 11.6	341.2 ± 15.8 #

* *p* < 0.001 (different from control), # *p* < 0.05 (different from diabetes + saline).

**Table 3 medicina-60-01580-t003:** Impact of diosmin and hesperidin on perineural thickness and plasma biomarkers in diabetic and control groups. Data are expressed as mean ± SEM.

	Control Group	Diabetes and Saline Treatment	Diabetes and Diosmin + Hesperidin 150 mg/kg Treatment	Diabetes and Diosmin + Hesperidin 300 mg/kg Treatment
Perineural Thickness (µm)	3.2 ± 0.5	17.3 ± 2.8 *	11.5 ± 0.9 #	7.9 ± 1.6 ##
Plasma FGF21 (pg/mL)	82.2 ± 9.1	106.2 ± 7.7 *	133.7 ± 14.8 #	161.1 ± 11.2 ##
Plasma Galectin-3 (ng/mL)	5.6 ± 0.3	11.8 ± 1.05 **	8.6 ± 0.7 #	7.3 ± 0.2 #
Plasma MDA (nM)	58.5 ± 3.3	203.7 ± 7.3 **	161.2 ± 10.4 #	105.9 ± 12.1 #

* *p* < 0.01 (different from control), ** *p* < 0.01, # *p* < 0.05, ## *p* < 0.01 (different from diabetes + saline).

## Data Availability

The datasets generated during and/or analyzed during the current study are available from the corresponding author on reasonable request.

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
