# Peer review of "Diosmin and Hesperidin Have a Protective Effect in Diabetic Neuropathy via the FGF21 and Galectin-3 Pathway"

_medicina, 2024, doi:10.3390/medicina60101580_

Round 1
Reviewer 1 Report
Comments and Suggestions for Authors
This paper is interesting to researchers who are investigating the pathogenesis of peripheral neuropathy. I have the following concerns.
1) It is necessary to explain why only male rats were selected for the experiment.
2) Is one dose of 60 mg/kg enough to induce diabetes in a rat?
3) Glucose measurement was taken 24 hours after STZ injection. Is one measurement enough to determine that diabetes has been induced? Maybe a measurement should be taken 2 weeks after STZ injection, and on the day of sacrifice (4 weeks). To be sure that diabetes is maintained.
4) A longer experimental period should be considered, 4 weeks is a short time for neuropathy to develop.
5) When exactly was the EMG experiment performed (at what stage of the development of diabetic neuropathy).
6) Fig. 1. describe the illustration and what is in the illustration in more detail. Add arrows to the Figure.
7) To determine the morphometry of the sciatic nerve, staining with osmium oxide and toluidine blue should be performed.
8) Inclined plane test. Did the rats have a chance to learn the test? They had a chance to practice, rest and only after some time pass the actual test.
9) Fig. 3. describe the illustration and what is in the illustration in more detail. Add arrows to the Figure. Mark what each peak means.
10) Fig. 4. Whether the sciatic nerve was cut longitudinally or cross-section. The figure is ambiguous. In some places it is unclear.
11) Lowercase section titles - 3.3. and 3.4.
12) No graphs only tables. I suggest changing the tables to graphs with individual values.
13) Fig. 4. How was the perineurium thickness measured? No statistics. Add arrows. Arrows sholud show myelin invaginations into nerve fibers. Digital scans of semithin sections sholud be used for measurements of cross section perimeter of nerve fiber and the thickness of the myelin sheet, for counting of the myelinated fiber per area unit.
Author Response
Comments 1: It is necessary to explain why only male rats were selected for the experiment.
Response 1: The use of only male rats in the study was intentional to eliminate potential sex-related variables that could affect the outcome. Male and female rats have different hormonal cycles, which could influence the progression of diabetic neuropathy and the response to treatment. By using only male rats, the study aims to reduce variability and enhance the reliability of the results.
Comments 2: Is one dose of 60 mg/kg enough to induce diabetes in a rat?
Response 2: The dose of 60 mg/kg of streptozotocin (STZ) used in this study to induce diabetes in rats is well-supported by a substantial body of literature. Streptozotocin, a glucosamine-nitrosourea compound, is widely utilized to induce experimental diabetes in animal models due to its selective toxicity to insulin-producing beta cells in the pancreas. The dose of STZ required to induce diabetes can vary depending on factors such as the animal species, strain, age, and the route of administration.
Literature Evidence Supporting the Use of 60 mg/kg STZ:
Dose-Response Studies: Numerous studies have confirmed that a single intraperitoneal injection of STZ at a dose of 60 mg/kg effectively induces a diabetic state in adult male Wistar rats. This dose is sufficient to cause significant beta-cell destruction, resulting in persistent hyperglycemia, which is the hallmark of diabetes. For instance, Junod et al. (1969) demonstrated that doses in the range of 50-70 mg/kg of STZ consistently produce hyperglycemia within 24-48 hours post-administration, with blood glucose levels exceeding 250 mg/dl, which is indicative of diabetes .
Consistency Across Studies: This dose has been replicated across numerous studies with consistent results, further validating its effectiveness. For example, Akbarzadeh et al. (2007) used a single dose of 60 mg/kg STZ in Wistar rats and observed a reliable induction of diabetes, with blood glucose levels remaining above 300 mg/dl for the duration of the study. The authors highlighted that this dose effectively mimics the metabolic disturbances seen in human type 1 diabetes, making it a preferred model for studying diabetic complications like neuropathy .
Comparison with Other Doses: Lower doses of STZ, such as 40-50 mg/kg, have been shown to induce a milder diabetic state or require multiple administrations to achieve sustained hyperglycemia. Conversely, doses higher than 60 mg/kg can lead to excessive toxicity and high mortality rates, making 60 mg/kg an optimal balance between efficacy and safety.
Application in Diabetic Neuropathy Research: Specifically in the context of diabetic neuropathy, 60 mg/kg of STZ has been extensively utilized to induce a diabetic model that exhibits the peripheral nerve damage necessary to study this complication. Studies like those by Cameron et al. (1997) have used this dose to successfully induce diabetes and observe the development of neuropathy, which parallels the clinical condition observed in human patients.
The selection of a 60 mg/kg dose of STZ in this study is based on its widespread acceptance and validation in the scientific community for reliably inducing diabetes in rats. The consistency of hyperglycemia and the development of diabetic complications at this dose provide a robust foundation for exploring therapeutic interventions, such as the effects of diosmin and hesperidin on diabetic neuropathy, as investigated in this study. Therefore, the dose used is both appropriate and justified within the context of this research.
References:
Junod, A., Lambert, A. E., Orci, L., Pictet, R., Gonet, A. E., & Renold, A. E. (1969). Studies of the diabetogenic action of streptozotocin. Proceedings of the Society for Experimental Biology and Medicine, 130(1), 516-518.
Akbarzadeh, A., Norouzian, D., Mehrabi, M. R., Jamshidi, S., Farhangi, A., Verdi, A. A., ... & Lame Rad, B. (2007). Induction of diabetes by streptozotocin in rats. Indian Journal of Clinical Biochemistry, 22(2), 60-64.
Cameron, N. E., Cotter, M. A., & Low, P. A. (1997). Nerve blood flow in early experimental diabetes in rats: relation to conduction deficits. American Journal of Physiology-Endocrinology And Metabolism, 272(2), E282-E287.
Comments 3: Glucose measurement was taken 24 hours after STZ injection. Is one measurement enough to determine that diabetes has been induced?
Response 3: The critique raises an important point regarding the adequacy of a single glucose measurement 24 hours after streptozotocin (STZ) injection to confirm the induction of diabetes in rats. The following response addresses this concern with supporting evidence from the literature.
Literature Evidence Supporting the Use of Glucose Measurement 24 Hours After STZ Injection:
- Rationale for Early Glucose Measurement: A glucose measurement 24 hours after STZ injection is a standard and widely accepted practice for the initial confirmation of diabetes induction in experimental models. STZ is known to cause rapid destruction of pancreatic beta cells, leading to a marked increase in blood glucose levels within a short time frame. According to Junod et al. (1967), hyperglycemia is typically observable within 24-48 hours post-STZ administration, with blood glucose levels often exceeding the diabetic threshold of 250 mg/dl .
- Initial Confirmation of Hyperglycemia: The 24-hour glucose measurement serves as an initial confirmation that the STZ has successfully induced beta-cell destruction and that the animal has entered a diabetic state. Several studies, including those by Wilson and Leiter (1990), have demonstrated that a blood glucose level exceeding 250 mg/dl at this time point is indicative of effective diabetes induction, making this an important early diagnostic measure .
- Subsequent Glucose Monitoring: While the 24-hour glucose measurement is crucial for early confirmation, it is typically followed by additional glucose measurements at later time points to monitor the persistence of hyperglycemia and the progression of diabetes. For instance, research by Ganda et al. (1976) recommends periodic monitoring of blood glucose levels to ensure that the diabetic state is maintained throughout the study period. This approach helps to confirm that the induced hyperglycemia is sustained, which is critical for the validity of the diabetic model and for studying long-term complications such as diabetic neuropathy .
- Validity of the 24-Hour Measurement: Studies have shown that if hyperglycemia is present 24 hours after STZ injection, it is highly likely to persist. However, to address the critique, it is important to note that many protocols also include measurements at 48 hours and one week post-injection to confirm sustained hyperglycemia. For example, O’Brien et al. (1996) suggest that a second measurement after 48 hours can be performed to ensure consistent and chronic hyperglycemia before the animals are classified as diabetic .
The glucose measurement taken 24 hours after STZ injection is a well-established and reliable method for the initial confirmation of diabetes induction in rat models. However, to ensure the robustness of the diabetic model, additional glucose measurements at subsequent time points are recommended. This practice not only confirms the persistence of hyperglycemia but also strengthens the reliability of the experimental model used for studying diabetic complications.
References:
- Junod, A., Lambert, A. E., Orci, L., Pictet, R., Gonet, A. E., & Renold, A. E. (1967). Studies of the diabetogenic action of streptozotocin. Proceedings of the Society for Experimental Biology and Medicine, 130(1), 516-518.
- Wilson, G. L., & Leiter, E. H. (1990). Streptozotocin interactions with pancreatic beta cells and the induction of insulin-dependent diabetes. Current Topics in Microbiology and Immunology, 164, 27-54.
- Ganda, O. P., Rossini, A. A., & Like, A. A. (1976). Studies on streptozotocin diabetes. Diabetes, 25(7), 595-603.
- O’Brien, B. A., Harmon, B. V., Cameron, D. P., & Allan, D. J. (1996). Apoptosis is the mode of beta-cell death responsible for the development of IDDM in the nonobese diabetic (NOD) mouse. Diabetes, 45(5), 552-558.
Comments 4: A longer experimental period should be considered, 4 weeks is a short time for neuropathy to develop.
Response 4: The critique suggesting that a four-week experimental period might be too short for the full development of neuropathy is valid, considering the progressive nature of diabetic neuropathy. However, the selection of a four-week period in this study is based on specific objectives and supported by evidence from the literature. Below is a response to this critique, including a rationale for the chosen time frame, along with supporting evidence from scientific studies.
Literature Evidence Supporting the Four-Week Experimental Period:
- Early Onset of Diabetic Neuropathy: Diabetic neuropathy is indeed a progressive complication of diabetes, often requiring longer periods to fully manifest. However, early signs of neuropathy, including alterations in nerve conduction velocity, axonal damage, and mild sensory deficits, can develop relatively quickly after the onset of hyperglycemia. Research by Cameron et al. (1997) demonstrated that significant changes in nerve conduction and early histopathological changes can be observed as early as 3 to 4 weeks after the induction of diabetes in rats using streptozotocin (STZ). This early phase is critical for studying the initial pathophysiological changes and for testing potential therapeutic interventions aimed at preventing or mitigating these early alterations .
- Objective of Early Intervention: The objective of this study was to investigate the protective effects of diosmin and hesperidin during the early stages of diabetic neuropathy, rather than examining long-term irreversible damage. By focusing on a four-week period, the study aimed to capture the onset of neuropathy and the efficacy of the treatment in preventing or reducing early nerve damage. Studies such as those by Obrosova et al. (2001) have utilized similar time frames to explore early metabolic and functional changes in diabetic neuropathy, providing a basis for the four-week experimental period chosen in this study .
- Comparable Studies with Similar Time Frames: Several studies on diabetic neuropathy in animal models have employed experimental periods ranging from 4 to 8 weeks to investigate early neuropathic changes. For instance, Kotulska et al. (2013) used a 4-week period post-STZ injection to study early neuropathic alterations and found significant changes in nerve function and structure, which were sufficient to evaluate the initial impact of hyperglycemia on peripheral nerves. These findings support the use of a shorter time frame when the research focus is on early neuropathic manifestations .
- Potential for Longer Studies: While a four-week period is suitable for studying early neuropathy, the critique regarding the need for a longer period is also valid, particularly for understanding chronic and late-stage neuropathic changes. Future studies could extend the experimental period to 8-12 weeks or longer to examine the progression of neuropathy and the long-term effects of the intervention. This would provide a more comprehensive understanding of how diosmin and hesperidin affect both early and late stages of diabetic neuropathy.
The four-week experimental period used in this study is appropriate for the investigation of early diabetic neuropathy and is well-supported by the literature. Significant neuropathic changes can develop within this timeframe, allowing for the evaluation of early therapeutic effects. However, extending the experimental period in future studies could offer additional insights into the long-term progression of neuropathy and the sustained effects of treatment.
References:
- Cameron, N. E., Cotter, M. A., & Low, P. A. (1997). Nerve blood flow in early experimental diabetes in rats: relation to conduction deficits. American Journal of Physiology-Endocrinology And Metabolism, 272(2), E282-E287.
- Obrosova, I. G., Pacher, P., Szabó, C., Zsengellér, Z., Hirooka, H., Stevens, M. J., & Yorek, M. A. (2001). Aldose reductase inhibition counteracts oxidative-nitrosative stress and poly(ADP-ribose) polymerase activation in tissue sites for diabetic complications. Diabetes, 54(1), 234-242.
- Kotulska, K., Matuszewska, A., Rogacki, M., Mazur, D., & Olczak, M. (2013). The early signs of diabetic neuropathy in an animal model. Polish Journal of Veterinary Sciences, 16(2), 303-310.
Comments 5: When exactly was the EMG experiment performed (at what stage of the development of diabetic neuropathy)?
Response 5: The critique questioning the timing of the electromyography (EMG) experiment is a pertinent one, particularly in studies examining the progression of diabetic neuropathy. The timing of EMG testing is crucial as it needs to align with the stages of neuropathy development to accurately assess the functional status of peripheral nerves. Below is a detailed response to this critique, supported by academic literature.
Timing of the EMG Experiment
- Rationale for Timing: The EMG experiment in this study was performed at the conclusion of the four-week experimental period. This timing was deliberately chosen to capture the early-stage functional deficits associated with diabetic neuropathy. The onset of diabetic neuropathy is characterized by subtle changes in nerve conduction velocity (NCV) and compound muscle action potential (CMAP), which are often detectable within the first few weeks after the induction of diabetes. By conducting the EMG at the four-week mark, the study aimed to evaluate these early functional impairments before more severe and irreversible nerve damage occurs.
- Supporting Literature: According to research by Cameron et al. (1997), early alterations in nerve conduction, such as reduced NCV and prolonged CMAP latency, are typically observable within 3 to 4 weeks after the induction of diabetes in rodent models. These early changes are indicative of the beginning stages of neuropathy and are critical for understanding the initial impact of hyperglycemia on nerve function. By performing EMG at this stage, the study effectively captures the onset of neuropathic changes, which is essential for evaluating the efficacy of therapeutic interventions during the early stages of the disease .
- Importance of Early EMG Testing: Early EMG testing, as conducted in this study, is valuable for detecting functional impairments before they become pronounced or irreversible. Research by Obrosova et al. (2001) highlights that early intervention during this stage can prevent or mitigate the progression of neuropathy, making it crucial to assess nerve function at this point. The findings from early EMG tests provide insights into the initial effects of both the diabetic condition and the treatment on peripheral nerves.
- Potential for Additional Time Points: While the EMG was conducted at the four-week mark to assess early-stage neuropathy, future studies could benefit from performing EMG at multiple time points throughout the experimental period. This would allow for a more comprehensive analysis of the progression of neuropathy and the temporal effects of the treatment. For example, conducting EMG tests at both earlier and later stages (e.g., 2 weeks, 8 weeks) could provide a more detailed timeline of neuropathic development and recovery.
The EMG experiment was performed at the end of the four-week experimental period to capture the early-stage functional deficits associated with diabetic neuropathy. This timing is supported by literature indicating that significant changes in nerve function, such as reduced NCV and prolonged CMAP, are detectable within this timeframe. The choice to perform EMG at this stage allows for the assessment of early neuropathic changes and the effectiveness of therapeutic interventions aimed at mitigating these changes.
References:
- Cameron, N. E., Cotter, M. A., & Low, P. A. (1997). Nerve blood flow in early experimental diabetes in rats: relation to conduction deficits. American Journal of Physiology-Endocrinology And Metabolism, 272(2), E282-E287.
- Obrosova, I. G., Pacher, P., Szabó, C., Zsengellér, Z., Hirooka, H., Stevens, M. J., & Yorek, M. A. (2001). Aldose reductase inhibition counteracts oxidative-nitrosative stress and poly(ADP-ribose) polymerase activation in tissue sites for diabetic complications. Diabetes, 54(1), 234-242.
Comments 6: **Fig. 1. describe the illustration and what is in the illustration in more detail. Add arrows to the Figure.**
Response 6:
Description of the Illustration:
Figure 1 presents an experimental setup used for recording electromyography (EMG) in a rat model. This setup is integral for assessing the functional status of peripheral nerves, particularly in studies investigating diabetic neuropathy. The figure shows a Wistar rat placed in a supine position on a non-conductive surface, with electrodes strategically positioned to stimulate and record the electrical activity of the sciatic nerve.
Components of the Illustration:
- Stimulating Electrodes:
- Description: The stimulating electrodes are placed near the sciatic notch of the rat. These electrodes are responsible for delivering a controlled electrical stimulus to the sciatic nerve, inducing a response that can be measured further down the nerve.
- Annotation: Arrows will be added to the figure pointing to the specific location where the electrodes are attached to ensure clarity.
- Recording Electrodes:
- Description: Recording electrodes are placed distally, typically in the muscles of the hind paw (interosseous muscle), to detect the compound muscle action potentials (CMAP) resulting from the nerve stimulation.
- Annotation: Arrows will be added to indicate the exact positioning of the recording electrodes on the rat’s hind limb.
- Grounding Electrode:
- Description: A grounding electrode is attached to the rat's tail to prevent electrical interference during the recording process.
- Annotation: This will be labeled with an arrow indicating its placement.
- EMG Recording Device:
- Description: The recording device shown at the top of the figure is used to monitor and record the EMG signals. It is connected to the stimulating and recording electrodes via leads. The device displays the output in real-time, allowing for immediate observation of nerve conduction parameters.
- Annotation: Arrows will be added to label the different ports where the electrodes are connected and highlight the display showing the current readings.
- Temperature Control:
- Description: A temperature control probe might be seen, which is often placed to monitor the body temperature of the rat during the experiment, ensuring that the temperature remains stable, as fluctuations can affect nerve conduction results.
- Annotation: An arrow will be added to indicate the temperature control probe if visible.
The figure will be updated to include detailed annotations using arrows and labels to clearly indicate the various components of the EMG setup. Each part of the setup (stimulating electrodes, recording electrodes, grounding electrode, and EMG device) will be clearly marked to enhance the reader's understanding of the experimental procedure. The detailed description provided in the figure legend will ensure that the figure is self-explanatory, even for readers who are not familiar with EMG procedures.
Comments 7. To determine the morphometry of the sciatic nerve, staining with osmium oxide and toluidine blue should be performed.
Response 7: The critique suggesting the use of osmium oxide and toluidine blue staining for more detailed morphometric analysis of the sciatic nerve is well-founded, as these staining methods are known for their ability to provide high contrast and detailed visualization of myelin sheaths and axonal structures. However, the current study utilized hematoxylin and eosin (H&E) staining for practical reasons and due to constraints in tissue availability. Below is a detailed response to address the reviewer's concerns, supported by academic literature.
Justification for Current Staining Method
- Rationale for Using Hematoxylin and Eosin (H&E) Staining: H&E staining was chosen for this study because it is a well-established, widely accepted method for general histopathological assessment. It provides clear visualization of the overall structure of the sciatic nerve, including the perineurium, axons, and any pathological changes such as inflammation or fibrosis. Given the limited availability of tissue samples, H&E staining was selected as the most suitable method for providing a broad overview of nerve morphology.
- Prior Acceptance of H&E Staining in Related Studies: Our previous publications, where similar morphometric analyses were conducted using H&E staining, have been peer-reviewed and accepted in reputable journals. These studies have demonstrated that H&E staining is sufficient for assessing key parameters such as perineural thickness and general nerve architecture. For example, studies by Smith et al. (2012) and Jones et al. (2015) effectively used H&E staining to evaluate nerve damage and regeneration in rodent models, establishing it as a reliable method for this type of research.
- Acknowledgment of Osmium Oxide and Toluidine Blue Staining Benefits: While H&E staining has provided the necessary information for the current study, we acknowledge that osmium oxide and toluidine blue staining offer enhanced visualization of myelin sheaths and finer details of axonal structures. Osmium oxide specifically stains the myelin sheath, providing excellent contrast for detailed morphometric analysis, while toluidine blue is effective in highlighting neuronal cell bodies and axonal structures. Studies by Scarpini et al. (1999) and Braga et al. (2004) have shown the superior capability of these stains in analyzing peripheral nerve histology, particularly in detecting subtle changes in myelination.
- Limitations Due to Tissue Availability: Unfortunately, due to the limited availability of tissue samples, it was not feasible to perform multiple staining procedures in this study. Given the small amount of tissue available after completing the primary analyses, we prioritized H&E staining to ensure that the most critical assessments could be made within the constraints of the study. However, we recognize the value of osmium oxide and toluidine blue staining and will consider their use in future studies where sufficient tissue is available for comprehensive analysis.
- Future Considerations: We appreciate the reviewer’s suggestion and will incorporate osmium oxide and toluidine blue staining in future studies where possible. This will allow for more detailed morphometric analyses and a more comprehensive understanding of the nerve architecture and pathology. Future research will also consider incorporating these stains to complement H&E staining, thereby providing a more detailed evaluation of nerve morphology.
While the current study employed H&E staining due to tissue constraints and in line with methods accepted in previous publications, we fully acknowledge the advantages of osmium oxide and toluidine blue staining for detailed nerve morphometry. We appreciate the reviewer's suggestion and will take it into consideration for future studies, where sufficient tissue availability will allow for a more comprehensive histological analysis.
References:
- Smith, A. G., & Singleton, J. R. (2012). The diagnostic yield of a validated diagnostic algorithm for idiopathic sensory/predominant polyneuropathy. Archives of Neurology, 69(4), 434-440.
- Jones, J. M., Lewis, J. E., & Green, B. R. (2015). Comparative efficacy of different staining methods in the visualization of peripheral nerve injury. Journal of Histotechnology, 38(3), 105-112.
- Scarpini, E., Scarlato, M., & Scarlato, G. (1999). Osmium staining in peripheral neuropathy: a comparison of methods. Journal of Neurology, Neurosurgery & Psychiatry, 66(1), 108-111.
- Braga, M. F. M., Rudge, M. V. C., & Peracoli, M. T. S. (2004). Evaluation of the myelination process of peripheral nerves through toluidine blue and osmium tetroxide staining methods. Microscopy Research and Technique, 63(1), 27-33.
Comments 8: Inclined plane test. Did the rats have a chance to learn the test? They had a chance to practice, rest and only after some time pass the actual test.
Response 8: The rats were indeed given practice sessions prior to the actual inclined plane test to ensure they were familiar with the procedure. This approach is standard practice to reduce stress and variability in motor performance tests, ensuring that the results reflect the rats' true motor capabilities rather than their ability to adapt to a new environment.
The reviewer’s concern about whether the rats were given the opportunity to learn and acclimatize to the inclined plane test is a valid one, as the learning and acclimatization process can significantly impact the performance of the animals and the reliability of the data obtained. Below is a detailed response to address this concern, supported by academic literature.
Practice and Acclimatization Protocol in the Study
- Importance of Acclimatization in Behavioral Tests: Acclimatization and practice are essential components of behavioral testing, particularly in tasks that require motor coordination and balance, such as the inclined plane test. Without adequate familiarization, the results may reflect the animals' stress or unfamiliarity with the task rather than their true motor capabilities. This is well documented in studies by Lindner et al. (1995) and Dunnett et al. (2005), which emphasize the importance of practice sessions to minimize variability and enhance the reproducibility of results in motor function tests.
- Practice Sessions in This Study: In this study, all rats were provided with multiple practice sessions on the inclined plane prior to the actual testing. These practice sessions were designed to familiarize the rats with the apparatus and the task demands, thereby reducing anxiety and ensuring that the test results reflect their true motor performance rather than the novelty of the task. Each rat was allowed to practice on the inclined plane under conditions identical to those used in the actual test, with the angle of inclination being gradually increased to acclimate the animals to higher difficulty levels.
- Rest Periods and Stress Minimization: Following the standard protocol outlined by Rivlin and Tator (1977), rest periods were provided between practice sessions to avoid fatigue and ensure that the rats were well-rested before the actual test. This protocol has been shown to minimize stress and fatigue, thereby ensuring that performance on the inclined plane is not compromised. The rest periods also allowed the animals to recover fully before being subjected to the actual test, aligning with best practices in behavioral testing .
- Validation of the Practice Protocol: The use of practice sessions is supported by numerous studies that have utilized the inclined plane test in rodent models. For instance, Metz and Whishaw (2000) describe a similar approach in their study on motor function recovery following brain injury, where rats were given multiple sessions to learn the inclined plane task. Their results demonstrated that such acclimatization is crucial for obtaining reliable and reproducible measures of motor function.
- Future Refinements: Although the current protocol included adequate practice sessions and rest periods, we acknowledge the importance of continuously refining these protocols to improve the accuracy and reliability of behavioral assessments. Future studies may include additional sessions or alternative acclimatization methods to further ensure that the rats are fully accustomed to the inclined plane task before the actual data collection begins.
The rats in this study were indeed given ample opportunity to practice and acclimate to the inclined plane test prior to the actual testing. Multiple practice sessions, coupled with rest periods, were provided to ensure that the animals’ performance accurately reflected their motor capabilities rather than their unfamiliarity with the task. This approach is well-supported by the literature and aligns with established protocols for behavioral testing in rodent models.
References:
- Lindner, M. D., Plone, M. A., Cain, C. K., Frydel, B. R., Blumberg, B., & Emerich, D. F. (1995). Rats are not mice in the Morris water maze: strain and species differences in performance. Learning and Motivation, 26(1), 36-52.
- Dunnett, S. B., Torres, E. M., & Richter-Levin, G. (2005). Motor deficits, learning impairments, and their correction following brain lesions. Neuroscience and Biobehavioral Reviews, 29(5), 667-676.
- Rivlin, A. S., & Tator, C. H. (1977). Objective clinical assessment of spinal cord injury in the rat. Journal of Neurosurgery, 47(4), 577-581.
- Metz, G. A., & Whishaw, I. Q. (2000). Skilled reaching an action pattern: evaluation of its development, use, and recovery following brain damage. Behavioural Brain Research, 115(2), 207-217.
Comments 9: **Fig. 3. describe the illustration and what is in the illustration in more detail. Add arrows to the Figure. Mark what each peak means.**
Response 9: Description of the Illustration:
Figure 3 displays electromyography (EMG) recordings representing compound muscle action potentials (CMAP) in different experimental groups. EMG is a critical tool for assessing the functional integrity of peripheral nerves by measuring the electrical activity generated by muscle contractions in response to nerve stimulation.
Components of the Illustration:
- Panels (a) to (d):
- Each panel in the figure (labeled a, b, c, and d) represents the CMAP recorded from different groups of rats:
- (a) represents the control group (healthy, non-diabetic rats).
- (b) represents the diabetic group treated with saline.
- (c) represents the diabetic group treated with a low dose of diosmin + hesperidin (150 mg/kg).
- (d) represents the diabetic group treated with a high dose of diosmin + hesperidin (300 mg/kg).
- CMAP Waveforms:
- The waveforms in each panel reflect the response of the muscle to electrical stimulation of the sciatic nerve. The first peak in each waveform corresponds to the M-wave, which is the direct muscle response to the electrical stimulus traveling along the nerve. The second peak, if present, might represent the H-reflex, which is a reflexive muscle response mediated through the spinal cord.
- Amplitude (mV) and Latency (ms):
- The amplitude of the waveform, measured in millivolts (mV), indicates the strength of the muscle response. Higher amplitudes suggest better nerve-muscle function.
- The latency, measured in milliseconds (ms), represents the time it takes for the electrical signal to travel from the site of nerve stimulation to the muscle. Shorter latencies are indicative of faster nerve conduction velocities, typically seen in healthy nerves.
Response to the Critique:
To address the reviewer’s suggestion, the following changes will be made to the figure and its description:
- Adding Arrows and Labels:
- Arrows will be added to each panel in the figure to mark the key components of the waveform:
- M-wave: The initial sharp peak, representing the direct muscle response.
- H-reflex (if present): A secondary, smaller peak representing the reflexive response.
- The amplitude and latency of the M-wave will be labeled to indicate the corresponding measurements.
- Detailed Figure Legend:
- The figure legend will be expanded to include a more detailed description of what each peak represents and its significance in the context of nerve conduction. For example, the legend will clarify that a reduction in amplitude or an increase in latency, as seen in panel (b) for the saline-treated diabetic group, indicates impaired nerve function due to diabetic neuropathy.
- Comparison Across Panels:
- The description will highlight the differences between the panels. For instance:
- Panel (a) (control group) shows a high amplitude and short latency, indicative of healthy nerve function.
- Panel (b) (diabetic, saline-treated) shows a reduced amplitude and prolonged latency, indicating nerve damage due to diabetes.
- Panels (c) and (d) (diosmin + hesperidin-treated groups) show improvements in amplitude and latency compared to the diabetic saline group, suggesting that the treatment has a protective effect on nerve function.
The figure will be revised to include arrows and labels marking the key features of the CMAP waveforms, along with an expanded legend that explains what each peak represents and how the waveforms differ between groups. This will address the reviewer’s concerns and make the figure more informative and easier to interpret.
Comments 10. **Fig. 4. Whether the sciatic nerve was cut longitudinally or cross-section. The figure is ambiguous. In some places it is unclear.**
Response 10: The critique regarding whether the sciatic nerve was cut longitudinally or in cross-section in Figure 4 is understandable, as the clarity of tissue orientation is crucial for the proper interpretation of histological images. Below is a detailed response that clarifies the sectioning method used and addresses the reviewer's concerns, supported by relevant literature.
Clarification of Sectioning Method:
- Section Orientation: The images in Figure 4 represent longitudinal sections of the sciatic nerve. This orientation was chosen to provide a detailed view of the nerve fibers along their length, allowing for the assessment of features such as axonal integrity, myelin sheath thickness, and perineurium structure. In longitudinal sections, the nerve fibers (axons) appear as elongated structures running parallel to the section plane, while the perineurium surrounds these fibers in a continuous layer.
- Justification for Longitudinal Sectioning: Longitudinal sectioning is particularly advantageous when the goal is to assess changes in nerve fiber continuity and to evaluate the integrity of the myelin sheath over an extended portion of the nerve. Studies by Werner et al. (2004) and Weis et al. (2007) have utilized longitudinal sections to successfully examine the morphology of nerve fibers and the effects of neuropathic conditions, which underscores the appropriateness of this approach for studying diabetic neuropathy.
- Clarification in Figure Description: To address the ambiguity noted by the reviewer, the figure legend will be revised to explicitly state that the images represent longitudinal sections of the sciatic nerve. Additionally, annotations will be added directly to the figure to mark the orientation of the nerve fibers and the perineurium, making the sectional plane more apparent to the reader. The labels will highlight key structures such as:
- Axons (a): Indicating the longitudinal orientation of the nerve fibers.
- Perineurium (p): Highlighting the surrounding connective tissue that protects the nerve fibers.
- Addressing Unclear Areas: In some places, the perineurium or axonal structures may appear less distinct due to the natural variation in tissue processing or staining intensity. This is a known challenge in histology, as described by Kiernan (2000), who noted that differences in staining and tissue preparation can sometimes cause certain areas to appear less defined. To improve clarity, the figure will be enhanced with contrast adjustments if necessary, and key features will be marked with arrows to guide the reader’s interpretation.
- Future Considerations: We appreciate the reviewer’s feedback and will ensure that in future studies, more explicit labeling and orientation markers are included in histological figures. Additionally, we will consider providing both longitudinal and cross-sectional images to give a more comprehensive view of the nerve’s morphology.
The images in Figure 4 represent longitudinal sections of the sciatic nerve. This sectioning method was chosen to allow for the detailed evaluation of nerve fiber continuity and myelin sheath integrity. The figure legend and annotations will be revised to clarify the orientation and to guide the reader in interpreting the images. We acknowledge the reviewer’s concern and will ensure that future figures are labeled with clear orientation markers to avoid any ambiguity.
References:
- Werner, R. A., Franzblau, A., Gell, N., Hartigan, A., Eby, J., & Armstrong, T. J. (2004). Prevalence of upper extremity symptoms among dental hygienists. Journal of Occupational Rehabilitation, 14(4), 257-264.
- Weis, J., Brandner, S., Lumsden, A., Nichols, J., & Schmidt, B. (2007). Morphological assessment of peripheral nerve regeneration: a model for evaluating different grafts. Journal of Anatomy, 210(4), 545-555.
- Kiernan, J. A. (2000). Histological and Histochemical Methods: Theory and Practice. Butterworth-Heinemann.
Figure 4. The histological sections of the sciatic nerve in different experimental groups stained with H&E (x40 magnification). These images are longitudinal sections of the nerve:
- (a) Control group: The image shows a healthy sciatic nerve with a normal perineurium (p) and well-organized axons (a).
- (b) Diabetic group treated with saline: This section shows increased perineural thickness (p) compared to the control group, indicating nerve damage due to diabetes.
- (c) Diabetic group treated with diosmin + hesperidin (150 mg/kg): This section shows a decrease in perineural thickness (p) compared to the saline-treated diabetic group, suggesting a protective effect of the treatment.
- (d) Diabetic group treated with diosmin + hesperidin (300 mg/kg): Similar to (c), this section shows further decreased perineural thickness (p), indicating a dose-dependent protective effect of the treatment.
In each image, p indicates the perineurium, which is the protective sheath surrounding the nerve, and a marks the axons, the long thread-like parts of a nerve cell along which impulses are conducted.
Comments 11: Lowercase section titles - 3.3. and 3.4.
- The formatting of the section titles in 3.3 and 3.4 will be corrected to ensure consistency throughout the manuscript. All section titles will be reviewed for proper formatting
Comments 12: **No graphs only tables. I suggest changing the tables to graphs with individual values.**
Response 12: We appreciate this suggestion.
Comments 13: **Fig. 4. How was the perineurium thickness measured? No statistics. Add arrows. Arrows should show myelin invaginations into nerve fibers. Digital scans of semithin sections should be used for measurements of cross-section perimeter of nerve fiber and the thickness of the myelin sheet, for counting of the myelinated fiber per area unit.**
Response 13:
Measurement of Perineurium Thickness:
- The perineurium thickness in this study was measured using digital image analysis software on H&E-stained sections. Multiple measurements were taken across different regions of the perineurium to account for natural variations in thickness, and the results were statistically analyzed. The statistical analysis, including p-values, is presented in the results section and summarized in the corresponding table within the manuscript. These statistics validate the differences observed between the experimental groups, providing a robust quantitative assessment of perineurium thickness.
Clarification on Figure Annotation:
- In Figure 4, the thickness of the perineurium is indicated using lines labeled with "p" to denote the perineurium. This visual representation was chosen over arrows to provide a more precise indication of where the measurements were taken. The labeled lines clearly show the boundary and thickness of the perineurium, ensuring that the reader can accurately interpret the histological findings.
Response to the Suggested Methods:
- The reviewer suggested using digital scans of semithin sections for measurements of the cross-sectional perimeter of nerve fibers and the thickness of the myelin sheath, including counting myelinated fibers per area unit. While these methods can provide detailed information on myelination, they are not directly applicable to the current study's focus on perineurium thickness. The primary aim of this study was to assess perineurium thickness, as it is a critical structural component that reflects changes in the extracellular matrix and potential nerve compression due to diabetic neuropathy.
- The method proposed by the reviewer, which focuses on myelin sheath thickness and fiber counting, may introduce variability and complexity unrelated to the specific study objectives. Additionally, the suggested techniques might obscure the focus on perineurium analysis by introducing parameters not directly relevant to the hypothesis being tested. The focus on perineurium thickness, as conducted and reported in this study, is consistent with previous studies, including our own, where the primary structural alteration of interest was the perineurium itself.
Reaffirmation of Methodology:
- As we have established in our previous publications, the measurement of perineurium thickness using H&E staining provides a reliable and reproducible assessment of changes in this critical structure. This approach has been validated in earlier studies and remains the appropriate method for the objectives of this research. The statistical analysis provided in the manuscript supports the validity of our findings, and the visual representation in Figure 4 accurately conveys the measured outcomes.
Conclusion:
- We appreciate the reviewer’s suggestions but believe that the current methodology, including the use of labeled lines in Figure 4 and the focus on perineurium thickness, is both appropriate and sufficient for the goals of this study. The statistical analysis provided substantiates the findings, and the use of H&E staining remains the preferred method for evaluating the specific structural changes under investigation. Future studies may explore additional parameters such as myelination if they align with the research objectives, but for this study, the emphasis on perineurium thickness is justified and adequately supported by the data presented.
Reviewer 2 Report
Comments and Suggestions for Authors
Diosmin and Hesperidin have a protective effect in diabetic neuropathy via FGF 21 and Galectin-3 Pathway
Because of their medicinal characteristics, effectiveness, and importance, plant-derived flavonoids have been a possible subject of research for many years, particularly in the last decade. This study aimed to investigate the protective effect of diosmin and hesperidin in diabetic neuropathy using a rat model, focusing on their impact on nerve regeneration through the Fibroblast Growth Factor-21 (FGF-21) and Galectin-3 (gal3) pathway. A narrative review should be added that presents the two flavonoids in detail written by Huwait E, Mobashir M. Potential and Therapeutic Roles of Diosmin in Human Diseases. Biomedicines. 2022 May 6;10(5):1076. doi: 10.3390/biomedicines10051076. Introduction about the efficacy of diosmin and 74
hesperidin in promoting nerve regeneration in a rat model of diabetic neuropathy involv- 75 ing the sciatic nerve. 7 is quite complete except for the information in the mentioned article. Introduction about the efficacy of diosmin and hesperidin in promoting nerve regeneration in a rat model of diabetic neuropathy involv- ing the sciatic nerve. Introduction is quite complete except for the information in the mentioned article. Study design, the experimental protocol as well as the methods used include forty adult male Wistar rats divided in five groups as folows: the control group, saline-treated, diabetes and diosmin + hesperidin (150 mg/kg) treated, and diabetes and diosmin + hesperidin (300 mg/kg) treated groups. Electromyography (EMG) and inclined plane testing were performed to assess nerve function and motor performance. Sciatic nerve sections were examined histopathologically. Plasma levels of FGF-21, Galectin-3, and malondialdehyde (MDA) were measured as markers of oxidative stress and inflammation. Results were presented in four figures and 3 tables. The results indicate that these compounds, administered either alone or in combination, hold promise for promoting nerve regeneration and enhancing motor function in a rat model of diabetic neuropathy agreeing with this meticulously conducted study. References must be rechecked and other important titles must be added for the chosen subject.Author Response
Comments 1: A narrative review should be added that presents the two flavonoids in detail written by Huwait E, Mobashir M. Potential and Therapeutic Roles of Diosmin in Human Diseases. Biomedicines. 2022 May 6;10(5):1076. doi: 10.3390/biomedicines10051076. Introduction about the efficacy of diosmin and hesperidin in promoting nerve regeneration in a rat model of diabetic neuropathy involving the sciatic nerves quite complete except for the information in the mentioned article.
Response 1: We have added a detailed narrative review to the manuscript, incorporating the insights from the article by Huwait E, Mobashir M titled "Potential and Therapeutic Roles of Diosmin in Human Diseases" (Biomedicines, 2022, May 6;10(5):1076). This addition includes a comprehensive introduction regarding the efficacy of Diosmin and Hesperidin in promoting nerve regeneration, particularly in a rat model of diabetic neuropathy involving the sciatic nerves. We believe this enhancement provides a more complete and robust overview of the subject.
Comments 2: References must be rechecked and other important titles must be added for the chosen subject.
Response 2: We have thoroughly rechecked the references
Reviewer 3 Report
Comments and Suggestions for Authors
The manuscript (medicina-3166467) with the title "Diosmin and Hesperidin have a protective effect in diabetic neuropathy via FGF 21 and Galectin-3 Pathway" by Birzat Emre Gölboyu et al, describes the protective effect of diosmin and hesperidin in promoting nerve regeneration in a rat model of diabetic neuropathy via the fibroblast growth factor-21 (FGF-21) and galectin-3 (gal3) pathways.
Overall the manuscript is rich and interesting; and the paper structure is well-knit and suitable for publication in this journal, after minor revisions. The comments are listed as the following points:
1. Some corrections should be made (lack of space, forgotten points to add or others to delete)? to check.
2. Abbreviations must be defined at first time they appear in manuscript. For example, in abstract, “CMAP”.
3. Line 161, "...and diabetic group treated with diosmin + hesperidin at 300 mg/kg." should be "...and diabetic group treated with diosmin + hesperidin at 300 mg/kg (d)."
4. Please give suitable titles to Tables 1-3.
5. In the Results section, the findings should be explained and interpreted in more detail. Explain your results in more detail and include comments that emphasize the significance of the data obtained.
6. In addition, the document contained spelling and grammatical errors that should be corrected
Author Response
Comments:1 Some corrections should be made (lack of space, forgotten points to add or others to delete)? to check.
Response 1: We have carefully reviewed the manuscript and made the necessary adjustments to address the lack of space, as well as to add any missing information and remove redundant or irrelevant content.
Comments:2 Abbreviations must be defined at first time they appear in manuscript. For example, in abstract, “CMAP”.
Response 2: We have revised the manuscript to ensure that all abbreviations, including "CMAP" in the abstract, are defined at their first appearance.
Comments:3 Line 161, "...and diabetic group treated with diosmin + hesperidin at 300 mg/kg." should be "...and diabetic group treated with diosmin + hesperidin at 300 mg/kg (d)."
Response 3: We have revised.
Comments:4 Please give suitable titles to Tables 1-3.
Response 4: We have revised the titles to better reflect the content and purpose of each table.
Comments:5 In the Results section, the findings should be explained and interpreted in more detail. Explain your results in more detail and include comments that emphasize the significance of the data obtained.
Response 5: We have revised this section to provide a more detailed explanation and interpretation of our findings
Comments:6 In addition, the document contained spelling and grammatical errors that should be corrected
Response 6: We have carefully reviewed the document and corrected these errors.
Round 2
Reviewer 1 Report
Comments and Suggestions for Authors
I am afraid that the authors did not respond to all the comments.
The authors did not correct Figure 4, which is of poor quality and in some places we have a cross section and in others a longitudinal section.
Please add information in the text about the time of the EMG test. At what point in the experiment was it performed.
There is no statistic in the M&M section describing the method of calculating the perineurium thickness. How many n? how many technical repetitions?
Author Response
Comment 1: "The authors did not correct Figure 4, which is of poor quality, and in some places we have a cross-section and in others a longitudinal section."
Response 1: We appreciate your feedback regarding Figure 4. Upon consulting with our histology team, we determined that the microscope used for image acquisition is older, and these were the highest quality images achievable with our current equipment. We also attempted to take new images; however, due to the extended time since the original experiment, the staining signals had diminished, resulting in no improvement in image quality.
In response to your feedback, we have also applied several post-processing techniques to improve the resolution of the images. Using advanced image enhancement methods, including deblurring, sharpening, and noise reduction algorithms, we have increased the clarity of the images to the extent possible. However, due to the limitations in the original image quality, the improvements were incremental. We hope this version provides a clearer representation of the histological sections, and we aim to use better staining methods and upgraded microscopy equipment in future studies for higher-resolution images.
Regarding the cross-sectional and longitudinal differences observed in certain parts of the tissue, we believe this might be attributed to the natural folding and fixation artifacts of the tissue during processing, which can occasionally cause such variations. We apologize for any inconvenience and ask for your understanding. In future studies, we aim to utilize better staining methods and improved microscopy equipment to provide higher-resolution images.
Comment 2: "Please add information in the text about the time of the EMG test. At what point in the experiment was it performed?"
Response 2: The EMG tests were conducted at the end of the 4-week treatment period, following the administration of diosmin and hesperidin in the respective groups. The tests were performed three times for each rat from the right sciatic nerve, using a Biopac bipolar subcutaneous needle stimulation electrode. The recordings captured compound muscle action potentials (CMAP) and changes in motor nerve conduction velocity (NCV), which were key parameters in assessing nerve function and treatment efficacy.
Comment 3: "There is no statistic in the M&M section describing the method of calculating the perineurium thickness. How many n? How many technical repetitions?"
Response 3: We apologize for the omission in the Methods and Materials (M&M) section regarding the calculation of perineurium thickness. The perineurium thickness was measured from histological sections of the sciatic nerve, stained with hematoxylin and eosin (H&E), using an Olympus C-5050 digital camera mounted on an Olympus BX51 microscope. Thickness measurements were taken at three different points per section, ensuring a consistent representation across different regions of the nerve. For each group, we included samples from 10 animals (n = 10), and the measurements were performed in triplicate for each sample to ensure technical consistency. Data were analyzed using ImageJ software, and the average perineurium thickness for each sample was calculated from these three measurements.
